# Particle Phase State and Aerosol Liquid Water Greatly Impact Secondary Aerosol Formation: Insights into Phase Transition and Role in Haze Events

Xiangxinyue Meng[1], Zhijun Wu[1,2]*, Jingchuan Chen[1], Yanting Qiu[1], Taomou Zong[1], Mijung Song[3], Jiyi Lee[4], Min Hu[1,2]

State Key Joint Laboratory of Environmental Simulation and Pollution Control, International Joint Laboratory for Regional Pollution Control, College of Environmental Sciences and Engineering, Peking University, Beijing 100871, China
Collaborative Innovation Center of Atmospheric Environment and Equipment Technology, Nanjing University of Information Science and Technology, Nanjing 210044, China
Department of Earth and Environmental Sciences, Jeonbuk National University, Jeonju, Republic of Korea, 54896
Department of Environmental Science and Engineering, Ewha Womans University, Seoul, Republic of Korea, 03760

*Corresponding author: zhijunwu@pku.edu.cn

**Abstract.** The particle-phase state is crucial for reactive gas uptake, heterogeneous, and multiphase chemical reactions, thereby impacting secondary aerosol formation. This study provides valuable insights into the significance of particle-phase transition and aerosol liquid water (ALW) in particle mass growth during winter. Our findings reveal that particles predominantly exist as semi-solid or solid during clean winter days with ambient relative humidity (RH) below 30%. However, non-liquid to liquid phase transition occurs when the ALW mass fraction exceeds 15% (dry mass) at transition RH thresholds of 40-60%. During haze episodes, the transformation rates of sulfate and nitrate aerosols rapidly increase through phase transition and increased ALW by 48% and 11%, respectively, resulting in noticeable increases in secondary inorganic aerosols (SIA). The presence of abundant ALW, favored by elevated RH and higher proportion of SIA, facilitates the partitioning of water-soluble compounds from gas to particle phase, as well as heterogeneous and aqueous processes in liquid particles. This leads to

a substantial increase in the formation of secondary organic aerosols and elevated
aerosol oxidation. Consequently, the overall hygroscopicity parameters exhibit a
substantial enhancement with a mean value of 23%. These results highlight phase
transition as a key factor initiating the positive feedback loops between ALW and
secondary aerosol formation during haze episodes over the North China Plain. Accurate
predictions of secondary aerosol formation necessitate explicit consideration of the
particle-phase state in chemical transport models.

## 1 Introduction

Submicron particles are ubiquitous in the nature, having great impacts on climate, visibility, and human health (Shiraiwa et al., 2011;Ravishankara, 1997;Pöschl, 2005;Lelieveld et al., 2015;Seinfeld et al., 2016;Hu et al., 2021). Phase state, a key parameter of particles, plays profound roles in the mass transport of reactive molecules between the gas phase and the particle phase (Marshall et al., 2018;Shiraiwa et al., 2011). This, in turn, influences the gas-particle partitioning of semi-volatile materials (Shiraiwa et al., 2013;Li and Shiraiwa, 2019), multiphase reaction rates of chemical species (Zhang et al., 2018;Mu et al., 2018), and even the ice nucleating activities of organic aerosols (OA) (Murray et al., 2010;Knopf and Alpert, 2023). Aerosol liquid water (ALW) contributes a substantial fraction of the mass in sub-micrometer particles on a global basis (Nguyen et al., 2016). Atmospheric particles with the presence of condensed water serve as suspended vessels of multiphase chemical reactions, leading to significant impacts on secondary aerosol formation, particle size growth, and air quality (Wu et al., 2018;Hodas et al., 2014;Liu et al., 2019). Therefore, a comprehensive understanding of particle-phase state and ALW is crucial for better evaluation of the related environmental effects.

In the real atmosphere, the particle-phase state varies significantly among solid, semi-solid, and liquid under different conditions, which specifically influenced by ambient relative humidity (RH), temperature, and aerosol chemical composition. For example, the atmospheric particles in the tropical rainforest over central Amazonia, which primarily consisted of secondary organic aerosols (SOA) derived from oxidation of isoprene, were observed to be in liquid state at RH > 80% (Bateman et al., 2016), but more non-liquid particles occurred with the impact of anthropogenic pollutants (Bateman et al., 2017). Liu et al. (2019) reported that particles with high mass fraction of inorganics and high RH were prone to be liquid in a subtropical coastal megacity. However, non-liquid particles appeared at RH < 60% in Beijing (Liu et al., 2017). Moisture can drive an RH-induced glass transition in particles, leading to a liquid state

and a significant water uptake at high RH in the lower atmosphere (Mikhailov et al., 2009). Moreover, organic aerosols might be in solid state at upper tropospheric temperatures that below about 210 K (Koop et al., 2011). Therefore, the changing features of aerosol composition and ambient RH may alter the ALW and trigger the phase state variation. More studies are needed to clarify the relationship between aerosol composition, particle-phase state, and ALW.

After the implementation of "China's Action Plan for Air Pollution Prevention and Control" in 2013, emissions of primary particulate matter and several gaseous pollutants have greatly reduced. However, the contribution and proportion of secondary inorganic aerosols (SIA) and secondary organic aerosols (SOA) have become increasingly significant (Lei et al., 2021;Wang et al., 2021b), especially during haze episodes in winter. As mentioned, particles changes from solid to liquid with elevated RH conditions during heavy haze episodes (Liu et al., 2017). In liquid particles, the gas-particle mass transfer for reactive gases can be greatly facilitated due to increased diffusion coefficients, and the thermodynamic equilibrium of semi-volatile compounds may be impacted to contribute to secondary aerosol formation (Shiraiwa et al., 2011;Jia et al., 2023). A recent field study by Gkatzelis et al. (2021) pointed out that the gas-to-particle partitioning in liquid particles enhances the uptake of water-soluble gas compounds, resulting in a 15-25% contribution of SOA mass during particulate pollution in Beijing. Many studies have demonstrated that the abundant ALW and high RH condition can greatly impact secondary aerosol formation processes (Xu et al., 2017;Wang et al., 2021a;Gkatzelis et al., 2021). However, there is still a lack of understanding regarding the role of phase state variations in secondary particulate pollution. In this study, we conducted a one-month field campaign in Beijing during winter to investigate the relationship between particle-phase state, ALW, and the chemical and physical processes involved in haze formation.

## 2 Methodology

### 2.1 Instruments and Measurements

Field campaigns were conducted in Beijing from 15[th] December 2020 to 10[th] January 2021 at the Changping campus of Peking University (40°8'N, 116°6'E). A detailed description of the sampling site can be found in previous studies (Wang et al., 2020c). The instruments were situated in the air monitoring laboratory, located on the top floor of the main building. A weather station (Met One Instruments Inc., USA), a suite of automatic gas analyzers ($O_3$, $SO_2$, CO and $NO_x$) from Thermo Scientific and an Aerodyne Quadrupole Aerosol Chemical Speciation Monitor (Q-ACSM) were operated according to standard protocols (Ng et al., 2011) and necessary information as described in Text S1.

The particle rebound fraction ($f$) was measured using a modified three-arm impactor (Bateman et al., 2014) coupled to a condensation particle counter (CPC, model 3772, TSI Inc.) with a time interval of 3 minutes, as described in our previous work (Liu et al., 2017;Meng et al., 2021). The three-arm impactor consisted of three parallel impactors with different designs. One of the impactors had no plate, while the others had plate equipped with an uncoated plate and a grease-coated plate, respectively. The no-plate impactor provided the total throughput rate, while the solid surface of the uncoated plate let particle rebound, and the sticky surface of the grease-coated plate captured all particles that struck it. To measure the $f$, a valve system with three solenoids and two actuators was used to ensure that the particle populations passing through the three impactors were sequenced and were measured by the CPC. Thus, rebound fraction, $f$, was defined as:

$$f = \frac{N_2 - N_3}{N_1 - N_3} \tag{1}$$

where $N_1$ was the whole particle population, $N_2$ was the population of particles that did not strike plus the rebounded particles from the impaction plate, and $N_3$ was the

population of particles that did not strike the impaction plate. Prior to measurement, we
dried the particles to below 30% RH using a silica gel diffusion dryer. Then, 300 nm
mono-disperse particles were selected by a Differential Mobility Analyzer (DMA, TSI
model 3080). An RH adjustment system with two RH probes and a Nafion RH
conditioner was employed to measure the RH conditions (ambient RH and impactor
RH), as well as to adjust the impactor RH to match the real atmospheric RH. The
measured impactor RH rapidly reached the ambient RH within 1 second, exhibiting a
mean absolute error of 0.03. This swift regulation time is attributed to the real-time
feedback in the RH control system, coupled with the typically modest fluctuations in
ambient RH. Particles with a diameter of 300 nm, as selected for our study, rapidly
achieved equilibrium in the humidification process, since the timescale for water
diffusion into these particles is approximately 1 second, shorter than their residence
time of about 3 seconds within the system. This is detailed in Text S2 and illustrated in
Figure S1. Such conditions ensure that the measured particle rebound are representative
and accurate at the ambient RH. Weekly calibrations using standard ammonium sulfate
and daily flow check were conducted (Liu et al., 2021;Liu et al., 2019). Typically, $f <$
0.2 or 0.1 are referred to the completely phase transition from non-liquid to liquid state
(Pajunoja et al., 2016;Liu et al., 2017). In this study, we consider $f < 0.2$ in the case of
liquid state. The time series of $f$ with an initial time resolution is shown in Figure 1,
while the data presented in other figures are all displayed as hourly averages.
**2.2 Data Analysis**
The mass concentrations of organics, sulfate, nitrate, ammonium, and chloride in non-
refractory particles (NR-PM$_1$) were analyzed using the standard ACSM data analysis
software (v.1.5.10). A collection efficiency (CE) of 0.5 was applied to the dataset (Xu
et al., 2017;Matthew et al., 2008). Positive matrix factorization (PMF) was performed
on the organic mass spectra using the Igor Pro based PMF2.exe algorithm to resolve
primary organic aerosols (POA) and SOA factors. The data and error matrices were
pretreated following methods from previous studies (Zhang et al., 2011;Zhang et al.,
2017). The key diagnostic plots are provided in supplementary (Figure S2-S3).
The aerosol liquid water content contributed by inorganics (ALW$_{inorg}$) in PM$_1$ was
estimated using the ISORROPIA-II thermodynamic model (Fountoukis and Nenes,
2007) with input of aerosol chemical composition measured by Q-ACSM. The particles
were assumed to be in metastable state, and the reverse mode was used to calculate the
ALW$_{inorg}$ due to absence of gaseous HNO$_3$ and NH$_3$. Besides, ALW associated with
organics (ALW$_{org}$) was considered using a simplified equation of $k$-Köhler theory (Guo
et al., 2015;Petters and Kreidenweis, 2007):
$$ALW_{org} = V_{org} k_{org} \frac{a_w}{1-a_w},\qquad\qquad (2)$$
where V$_{org}$ is the volume concentration of organics with a typical density of 1.4 g/cm$^3$
(Cerully et al., 2015), $k_{org}$ is the hygroscopicity parameter of the organics, a$_w$ represents
the water activity, which is assumed to have the same value as RH. In this study, we
used a fixed $k_{org}$ of 0.06 to evaluate ALW$_{org}$, which was the average value of the overall
$k_{org}$ in the consideration of POA and SOA contributions in the total non-refractory
organics ($k_{POA}$ = 0 and $k_{SOA}$ = 0.1) (Wu et al., 2016;Gunthe et al., 2011). However, it
should be noted that $k_{org}$ has been found to exhibit a positive linear relationship with
the aerosol oxidation degree, which varied among species (Chang et al., 2010;Duplissy
et al., 2011). $f_{44}$, the fraction of m/z 44 fragment signal to total organic signal, is widely
used to represent the atmospheric aging process of OA species (Ng et al.,
2010;Canagaratna et al., 2015). Real-time $k_{org}$ was $k_{org} = 1.04 \times f_{44} - 0.02$, as
reported by Kuang et al. (2020) for the North China Plain (NCP). The predicted real-
time $k_{org}$ ranged from 0.13 to 0.24, which was consistent with the variation range
reported for winter Beijing (0.06-0.3) (Li et al., 2019;Jin et al., 2020). For fixed $k_{org}$,
the contribution of organics to ALW was ~12% on average during the observation.
However, considering the variation of real-time $f_{44}$, organics were capable to provide
more than 30% and 20% of the total ALW mass on average during clean and polluted
days, respectively (Figure S4 and Text S3). In recognition of ALW's plasticizing effect
on particle-phase state, the potential impacts of whether the hygroscopicity value of
organics is fixed or varies in real-time on phase transition has been discussed in Section

173 3.2.

For a given internal mixture, the overall particle hygroscopicity ($k_{total}$) was calculated
by a simple mixing rule by weighting the hygroscopicity parameters of the components
by their volume fractions in the mixture (Petters and Kreidenweis, 2007):
$$k_{total} = k_{inorg} \cdot frac_{inorg} + k_{org} \cdot frac_{org}, \qquad (3)$$
Where $frac_{inorg}$ and $frac_{org}$ are the inorganics and organics volume fractions in NR-PM$_1$,
respectively. Considering the variability in the composition of inorganics and organics,
the hygroscopicity parameters of inorganics ($k_{inorg}$) was weighted by volume fractions.
The main form of inorganic species (NH$_4$NO$_3$ and (NH$_4$)$_2$SO$_4$) in the urban atmosphere
was considered due to the lower abundance of chloride in NR-PM$_1$. The volume fraction
of each inorganic species was calculated based on the ion-pairing scheme as described
in Gysel et al. (2007) with their gravimetric density (1720 kg m$^{-3}$ for NH$_4$NO$_3$ and 1769
kg m$^{-3}$ for (NH$_4$)$_2$SO$_4$) (Wu et al., 2016). The hygroscopicity parameters of NH$_4$NO$_3$
and (NH$_4$)$_2$SO$_4$ are 0.58 and 0.48, respectively following previous studies (Wu et al.,
2016;Jin et al., 2020;Petters and Kreidenweis, 2007). For hygroscopicity parameters of
organics ($k_{org}$), real-time $k_{org}$ were used as above, which effectively captured the
characteristics of the investigated area in our study.
**3 Results and Discussion**
**3.1 Chemical Composition and Phase State of Sub-micrometer Particles**
Figure 1 shows the time series of meteorological parameters, chemical composition of
NR-PM$_1$, gas pollutants, and particle rebound fraction from December 16, 2020, to
January 10, 2021. The average mass concentration of NR-PM$_1$ was 15.8±16.8 μg m$^{-3}$
during the measurement period. During clean periods (NR-PM$_1$ < 20 μg m$^{-3}$), organics
dominated the aerosol composition, accounting for ~45% of NR-PM$_1$ mass. Nitrate,
sulfate, and ammonium contributed 20%, 16%, and 16% to total NR-PM$_1$ on average,
respectively (Figure S5). However, several pollution episodes occurred with rapid
growth in NR-PM$_1$ and ALW mass concentration with higher concentrations of NO$_x$
and SO$_2$, as marked by yellow shadow in Figure 1. These four polluted episodes
typically started with ambient RH below 40% and higher O$_3$ levels (> 30 ppb) and
mounted up with stagnant meteorological conditions bringing high RH (> 60% RH)
and low surface wind speed (< 3 m/s). This meteorological pattern is commonly
observed over the NCP during haze episodes (Sun et al., 2013;Sun et al., 2015). During
these polluted episodes, nitrate increased rapidly accounting for an average of 33% of
the total NR-PM$_1$ mass. ALW was minor during clean days, but increased up to 26% in
PM$_1$ during severe polluted episodes with NR-PM$_1$ > 80 μg m$^{-3}$ (Figure S6). The mass
concentrations of POA and SOA both increased during these polluted episodes as
shown in Figure 1d. Moreover, the mass contribution of SOA to total OA showed an
upward trend in particulate mass, indicating the important contribution of secondary
formation during haze formation (Figure S7).
As shown in Figure 1f, particle rebound fraction, $f$, varied with ambient RH from 1.0 to
~0.0 during the observation, indicating that particles possessed phase transition from
non-liquid to liquid state. Similar patterns of particle-phase transition were found for
several polluted episodes. Taking P4 as an example, $f$ remained stable at 0.8 with RH =
~20% during the initial period of stagnant conditions, but gradually dropped to ~0.1
along with the increasing RH and NR-PM$_1$ during the subsequent haze formation. In
addition, we collected several PM$_{2.5}$ filter samples to characterize the bulk-phase
viscosity during clean and polluted days based on poke-and-flow experiment, as
described in our previous study (Song et al., 2022) and Text S4 (indicated by black and
red frames in Figure 1f). As shown in Figure S8, the viscosity was proved to be higher
than ~10$^8$ Pa s with a mean value of $f$ > 0.8 during clean days, indicating that particles
existed in a solid or semi-solid state. However, the viscosity was lower than ~10$^2$ Pa s
with an average $f$ <0.2 under higher RH conditions during polluted days, indicating the
liquid state. It should be noted that the viscosity measurement captured the bulk-phase
viscosity for water soluble components in $PM_{2.5}$ filter samples, but the online
measurement of $f$ depicted 300 nm particles representative of accumulation mode
particles, which normally contributed the majority fraction of $PM_1$. While the
differences in chemical composition between $PM_{2.5}$ filter samples and 300 nm particles
may introduce uncertainties when comparing the phase state of the targeted aerosols,
the viscosity results showed good agreement with the average variation of $f$ during the
corresponding period. Further validation is still necessary to compare the two different
techniques and will be displayed in our further study. To directly indicate the phase
transition from the perspective of viscosity, RH-dependent $f$ was measured for these
filter samples with known bulk-phase viscosity (Figure S9 and Text S5). As expected,
the decreasing $f$ from >0.8 to 0 covered the transition range from ~$10^8$ Pa s to ~$10^2$ Pa
s, which indicated the consistent behavior of particle rebound and measured bulk-phase
viscosity for the investigated aerosols.
**3.2 Phase Transition Behavior of Sub-micrometer Particles**
Figure 2a illustrates the frequency distribution of RH. $f$ as a function of RH were plotted
in Figure 2b. During the observation, ambient RH was below 30% for more than half
of the time with $f$ predominantly exceeding 0.8 under such conditions. When RH
increased to ~50-60%, a majority of $f$ dropped to <0.2 along with the increasing NR-
$PM_1$ mass. This means that particles went through a moisture-induced phase transition
from non-liquid to liquid during haze formation when RH reached 60%, which aligned
with our previous studies (Liu et al., 2017). Notably, some points with higher mass
fraction of inorganics ($f_{inorg}$ > 0.7) showed $f < 0.2$ at RH = 40-50%, indicating that
particles with higher $f_{inorg}$ were already in a liquid state. Consequently, particles
underwent phase transition with a relatively large RH range of 40-60%, exhibiting
varying chemical compositions as marked by the red frame.
Particle-phase state is known to be sensitive to ALW by its unique plasticizer effect
(Koop et al., 2011). In Figure 2d, $f$ as a function of ALW/NR-PM$_1$ were plotted to
represent the relative water uptake of unit mass dry aerosols with corresponding particle
rebound behaviors. Figure 2c displays the frequency distribution of three $f$ intervals in
each ALW/NR-PM$_1$ bin. When ALW/NR-PM$_1$ < 5%, the frequency of $f$ > 0.8 was
higher than 0.65, indicating that particles mostly stay in a more viscous non-liquid state
with less water uptake capacity. When ALW/NR-PM$_1$ increased to 5-15%, $f$ gradually
decreased from 0.8 to 0.2, suggesting that the total water uptake gradually enhanced
and lowered the viscosity to trigger the phase transition within this range. The non-
liquid particles were dominant with the frequency of $f$=0.2-0.8 close to 0.8. When
ALW/NR-PM$_1$ > 15%, the frequency of $f$ < 0.2 dramatically increased from 0.2 to ~0.8,
reaching close to 1.0 at ALW/NR-PM$_1$ > 25% with higher particulate mass. This
indicates that particles mostly convert to liquid when the mass fraction of ALW
surpasses a certain threshold during haze formation, rather than the absolute ALW mass
(Figure S10). In general, a good correlation between ALW/NR-PM$_1$ and $f$ was observed.
ALW/NR-PM$_1$, used as a mass-based hygroscopic growth factor (Chen et al., 2022;Liu
et al., 2018), is suitable to quantify the moisture-induced phase transition capacity of
atmospheric particles, and a value of 15% can be the sudden change in the case of phase
transition from non-liquid to liquid.
It should be noted that calculations of ALW in this study have considered inorganic
salts and organics. Acknowledging that the hygroscopicity of organics, characterized
by either a fixed $k_{org}$ or varying in real-time, affects the calculation of ALW mass, a
sensitivity analysis examining its impact on the phase transition threshold (ALW/NR-
PM$_1$) is presented in Figure S11. There is no denying that the contribution of inorganic
salts to ALW remains predominant, with their contribution to ALW being ~88% (a fixed
$k_{org}$ of 0.06) and ~73.5% (real-time $k_{org}$) on average during the observation (Figure S4),
indicating that the impact of different ALW calculations on ALW/NR-PM$_1$ values was
minor. As expected, the frequency distribution of these three $f$ intervals showed no
obvious change for ALW calculations by inorganics, fixed $k_{org}$, and real-time $k_{org}$ at the
whole ALW/NR-PM$_1$ range. Although the frequency of $f < 0.2$ changed from ~0.8 to
0.5 at ALW/NR-PM$_1$ = 15-20% when shifting to real-time $k_{org}$, the frequency remained
higher than 0.5 and approached 1.0 with larger ALW/NR-PM$_1$ values. This indicates
that while the varying $k_{org}$ impacts the number of data points within ALW/NR-PM$_1$ bins
of 15-20%, it does not affect the overall transition trend. As a result, the impacts of
different ALW calculation on ALW/NR-PM$_1$ were not significant, and the phase
transition threshold of 15% remains valid. It is suggested that caution should be
exercised when using the above approach to characterize the phase state of targeted
aerosols, as the measured $f$ was representative of accumulation mode particles that
dominated the mass concentration of submicron particles (Seinfeld, 2006).
It is interesting to note that several points with ALW/NR-PM$_1$ < 5% and NR-PM$_1$ >30
μg/m$^3$ exhibited lower rebound fraction ($f < 0.4$), which was attributed to the variation
of RH background from high RH to low RH during the later stages of the haze episodes,
as shown in Figure 2d and Figure S12. This suggests that liquid particles may not turn
to be a more viscous semi-solid state in a brief period under dehydration process. There
are two possible explanations for this phenomenon. Firstly, the presence of significant
amounts of inorganic and organic compounds can alter the humidity conditions for
deliquescence and efflorescence (Ushijima et al., 2021;Peckhaus et al., 2012). Secondly,
these particles are likely become non-ideal mixing due to drying process that form core-
shell structure (Shiraiwa et al., 2013;Ciobanu et al., 2009;Song et al., 2013). Studies
have revealed that outer phase may form viscous organic shell to prevent water
evaporation (Koop et al., 2011;Shiraiwa et al., 2013;Hodas et al., 2015), thus, the inner
phase containing inorganics still keep liquid with residual water. However, it should be
noted that liquid-liquid phase separation was not optically detected under staged
dehydration of filter-based Beijing PM$_{2.5}$ droplets by Song et al. (2022). Instead, they
observed abrupt effloresced inorganics at ~30% RH, which was much lower than
(NH$_4$)$_2$SO$_4$ and NH$_4$NO$_3$ in pure form (Peng et al., 2022). This supports that
atmospheric particles are more likely to be metastable after liquification only if RH
decreases to very low values.
In addition to RH and aerosol compositions, environmental temperature also plays a
significant role in determining the phase state (Koop et al., 2011;Shiraiwa et al.,
2017;Petters et al., 2019). A reduction in temperature results in higher viscosity,
whereas a rise in RH leads to a decrease in viscosity, attributed to the plasticizing effect
of water (Koop et al., 2011). Although the relationship between $f$ and temperature is not
strongly evident as that of RH in this study (Figure S13), it's observed that a greater
number of data points exhibited near 0.9 under low RH conditions (<30%), suggesting
higher viscosity at colder temperatures (< -10 ℃) than warmer scenarios. The glass
transition temperature ($T_g$) is a key metric for the non-equilibrium phase transition from
a glassy solid to a semi-solid state as temperature rises (Koop et al., 2011). Particles act
as solid when the temperature falls below $T_g$ ($T_g/T>1$), and transition to semi-solid or
liquid at temperature exceeding $T_g$. An increase in compound molecular weight, O:C
ratio, and functional group composition are identified as key factors affecting the $T_g$ of
OA (Saukko et al., 2012;Dette et al., 2014;Rothfuss and Petters, 2017;Shiraiwa et al.,
2017). Shiraiwa et al (2017) proposed that $T_g/T$ is an indicator for the semi-solid to
liquid phase transition of OA, with a threshold of $T_g/T \approx 0.8$. In this study, we employed
a $T_g$ parameterization method for OA viscosity based on their molecular weight and
O:C ratio to assess the combined effects of aerosol composition, RH and temperature
on particle phase state (Shiraiwa et al., 2017). This method accounts for water
associated with both inorganics and organics, rather than focusing solely on organics,
to calculate $T_g$ of ambient OA, as elaborated in Text S6.
Figure 3a displays the characteristic relations between $T_g/T$ and ALW/NR-PM$_1$ with
different approach for $T_g$ calculations of ambient OA. Different phase state intervals are
characterized by $T_g/T$ based on predicted viscosity $\eta$ as shown in Figure 3b, and are
illustrated using dashed lines with arrows. The predicted viscosity $\eta$ of OA was
calculated by applying the Vogel–Tammann–Fulcher (VTF) equation (Angell, 1991)
with a fragility parameter of 10 (DeRieux et al., 2018). Clearly, after calculating $T_g$ in
conjunction with $k_{total}$, there is a strong consistency in the characteristic relationship
between the estimated $T_g/T$ and ALW/NR-PM$_1$ with both fixed and variable $k_{org}$. This
consistency aligns well with the phase state changes of atmospheric aerosols discussed
earlier in this study. In contrast, even when accounting for variations in hygroscopicity
due to different oxidation degrees of OA, the majority of these estimated $T_g/T$ values
fall within the semi-solid and solid range at higher ALW/NR-PM$_1$, significantly
deviating from the field observations. This highlights the significant impact of
environmental RH and chemical composition on the moisture-induced phase transition
of atmospheric particles in the near-surface atmosphere. In particular, inorganic salts
play a dominant role, contributing more significantly to the mass fraction of ALW in
total particulate matter. The estimated $T_g/T$ for ambient OA with $k_{total}$ transitioned to a
liquid state at ALW/NR-PM$_1$ > 10%, which is slightly lower than the transition
threshold of 15% proposed in this study. It should be noted that the estimated $T_g$ of OA
adopted an average molecular weight (MW) of 200 g mol$^{-1}$, as used in previous studies
(Williams et al., 2010;Shen et al., 2018). However, the average MW of ambient OA is
likely variable due to the atmospheric aging process. Increasing the value of MW can
shift the characteristic cure of $T_g/T$ versus ALW/NR-PM$_1$ to the right, thereby aligning
the semi-solid to liquid transition threshold more closely with the results observed in
this study. This further suggests that incorporating of $k_{total}$ into $T_g$ calculation may
potentially enhance the simulation results, especially in regions with a high proportion
of inorganic salts under humid conditions. It should be noted that this aspect warrants
further exploration in subsequent research.
**3.3 Effects of Phase Transition and ALW on SIA Formation during Haze Episodes**
We investigated the $f$ and secondary aerosols during four polluted episodes (P1 to P4)
under stagnant weather conditions with WS < 3 ms$^{-1}$. Sulfur and nitrogen oxidation
ratios, SOR ($nSO_4/(nSO_4 + nSO_2)$) and NOR ($nNO_3/(nNO_3 + nNO_2)$), commonly
used as indicators for secondary inorganic transformation (Li et al., 2017), are plotted
as a function of $f$ in Figure 4a and 4b. We found that SOR (NOR) remained in a lower
level with a mean value of ~0.27 (0.08) at $f > 0.2$ for non-liquid particles, but increased
significantly to ~0.8 (0.35) with increasing ALW/NR-PM$_1$ at $f < 0.2$. This indicates that
the secondary formation of SIA is facilitated to a certain degree through phase transition
and the increasingly higher ALW mass. It should be noted that particles can be non-
liquid during haze episodes with $f$ =1.0-0.2. Interestingly, SOR and NOR remained in
lower levels and did not show notable increase between $f$ = 1.0-0.8 and $f$ = 0.8-0.2, until
particles accomplished the phase transition at $f$ = 0.2-0.0 (Figure 4c1 and 4c2). As a
result, the median SOR (NOR) increased to higher levels with an increment of 48%
(11%) via phase transition along with the increase in ALW.
From the perspective of phase state, the increasing mass fraction of ALW reduces the
viscosity and triggers the phase transition, which have important roles in the gas-
particle mass transfer during haze formation. It is suggested that the secondary
transformation of SIA is impeded by limited mass transfer between gas and particle
phase when particles are not fully converted into liquid state. However, these limited
factors disappear or the dominant formation pathway changes after phase transition. As
reported in previous studies, ALW facilitates the secondary formation of sulfate and
nitrate via the promotion of heterogeneous reactions (e.g. $SO_2$ heterogeneous oxidation,
$N_2O_5$ hydrolysis), gas-particle partitioning of semi-volatile components or aqueous-
phase reactions on wet aerosols (Chen et al., 2022;Cheng et al., 2016;Wang et al.,
2020b;Liu et al., 2020). However, aqueous-phase oxidation of $SO_2$ may be constrained
before phase transition due to the low diffusivity of multiple oxidants (e.g. $O_3$, $H_2O_2$
and $NO_2$) in the particles, and it may become the dominant formation pathway in liquid
particles (Ravishankara, 1997;Liu et al., 2020). Additionally, the partitioning of nitrate
into particles following Henry's Law may also be facilitated by the increased ALW due
to enhanced diffusivity of dissolved precursors in liquid particles. In Figure 4d, the mass
fraction of SIA ($f_{SIA/NR-PM_1}$) is plotted as a function of $f$. The $f_{SIA/NR-PM_1}$, ALW mass
concentration, and RH were grouped and averaged corresponding to $f$ bin width of 10%.
We found that $f_{SIA/NR-PM_1}$ remained stable at $f$ = 1.0-0.4, but steadily increased from an
average of ~0.50 to ~0.65 with elevated RH levels (>40%) and decreasing $f$ (from 0.4
to 0.0). This indicates that SIA formation was limited for non-liquid particles with
higher viscosity under lower RH conditions. However, ALW was steadily enhanced by
the increasing RH and started to trigger the phase transition, thereby facilitating the
SOR and NOR to a larger extent. Therefore, $f_{SIA/NR-PM_1}$ apparently increased with the
increase in ALW at $f$ = 0.2-0.0. The presence of more ALW in liquid particles was
expected to promote the SIA formation by acting as multiphase reaction vessels (Zheng
et al., 2015;Wang et al., 2020a;Wang et al., 2020b). One should note that, the average
environmental temperature during pollution episodes increased to approximately 0°C,
in contrast to the -10°C recorded during clean periods. The rise in ambient temperature
typically enhances the diffusivity of atmospheric reactive molecules in both the gas and
particle phases (Tang et al., 2014;Shiraiwa et al., 2011;Li and Shiraiwa, 2019). This, in
turn, may potentially influence the heterogeneous or liquid-phase reactions, and even
the gas-particle partitioning of semi-volatile compounds.
**3.4 Effects of Phase Transition and ALW on SOA Formation during Haze Episodes**
In Figure 5a and 5b, the ratio of SOA to POA (SOA/POA) is plotted as a function of $f$
during these four polluted episodes characterized by $ALW/NR-PM_1$ and $f_{44}$. For $f$ = 1.0-
0.2, particles possessed relatively lower SOA/POA values (1-2.5) with $ALW/NR-PM_1$
<15%, which was independent of $NR-PM_1$ mass concentrations. However, a noticeable
increase in SOA/POA and elevated $f_{44}$ values were observed at $f$ = 0.2-0.0, accompanied
by increasing $ALW/NR-PM_1$ and $NR-PM_1$ mass. This indicates that more oxidized SOA
was produced in liquid particles through the phase transition and the increasing mass
fraction of ALW during haze formation. Interestingly, we observed that these liquid
particles were primarily associated with polluted days during the nighttime (Figure
S14). For these liquid particles, SOA/POA doubled to ~5.5 along with the increasing
$f_{44}$ compared to non-liquid particles, suggesting the important roles of phase transition
and ALW in promoting the SOA formation through dark reactions during nighttime.
From the perspective of phase state, phase transition was directly indicated by the
decreasing $f$ during haze formation driving a large decrease in bulk phase viscosity
from $>10^8$ Pa s to $<10^2$ Pa s as proved by viscosity measurement, which may enhance
the gas-particle mass transfer. ALW reduces the viscosity and triggers the phase
transition, thus facilitating the uptake of precursors and oxidants, and potentially
altering the reaction pathway (Tillmann et al., 2010;Berkemeier et al., 2016;Li et al.,
2018;Zhao et al., 2019).
For non-liquid particles, ALW facilitates the SOA formation via partition and
heterogeneous uptake of water-soluble organics from gas phase into the particle phase,
leading to a rapid increase in SOA along with ALW (Herrmann et al., 2015;Gkatzelis
et al., 2021;Lim et al., 2010;El-Sayed et al., 2015). Subsequent aqueous-phase reactions
may occur to form oligomers, organosulfates, and nitrogen-containing organics through
radical or non-radical reactions (Surratt et al., 2007;Iinuma et al., 2007;Galloway et al.,
2009;Lim et al., 2013;Wang et al., 2020c). However, these reactions may be limited in
non-liquid particles by the lower diffusivity due to higher viscosity. In contrast, liquid
particles provide unstrained mass transfer of necessary oxidants and precursors between
gas and particle phase, which is favorable for aqueous-phase processing. It is well
known that aqueous-phase processing can contribute more oxidized SOA (Xu et al.,
2017;Ervens et al., 2011;Zheng et al., 2023). Recent field studies have demonstrated
that oligomers or dicarboxylic acids were enriched in liquid particles from the reactive
uptake of methylglyoxal during the severe haze episodes in Beijing (Zheng et al., 2021).
These oxidation products formed through aqueous-phase reactions are typically more
oxidized and less volatile than those formed through gas phase photochemistry (Ervens
et al., 2011), which can be reserved in the particle phase and increased the SOA mass
in total OA. Therefore, the significant growth of SOA/POA and $f_{44}$ after phase transition
is attributed by the enhanced heterogeneous or aqueous-phase reactions in liquid
particles with abundant ALW during the nighttime.

**3.5 Positive Feedback Loops between ALW and Secondary Aerosol Formation Triggered by Phase Transition during Haze Episodes**

In Figure 6a, the relationship between the overall particle hygroscopicity($k_{total}$) and RH is displayed. The $k_{total}$, ALW, and NR-PM$_1$ mass were grouped and averaged corresponding to an RH bin width of 10%. When RH was below 30%, the averaged $k_{total}$ was ~0.35. However, it increased to 0.39 with higher ALW and NR-PM$_1$ mass at RH =40-60%, and further rose to 0.43 with an average maximum NR-PM$_1$ value of 56 μg/m$^3$ when RH reached 70-80%. This indicates that hygroscopic growth of particulate matter underwent two stages with increasing RH and NR-PM$_1$ mass, particularly at RH = 40-60% and RH > 70%. From the above discussion, we have demonstrated that the non-liquid to liquid phase transition was triggered by the increased ALW, with a transition RH threshold of 40-60% during haze episodes (as indicated by gradual color change in Figure 6a). Phase transition facilitated the formation of sulfate and nitrate aerosols, contributing higher proportion of SIA in total particles under higher RH conditions. Notably, this led to a continuous increase in the volume fraction of inorganics with increasing RH (Figure 6b). Besides, $k_{inorg}$ also slightly increased when RH reached 60% due to increased nitrate contribution in total SIA during haze episodes (Figure 6c and Figure S5). This may explain the first enhancement of $k_{total}$ at RH = 40-60%, which was mainly driven by the large increase in $frac_{inorg}$ favored by phase transition.

Furthermore, the increase in $k_{total}$, coupled with elevated RH levels, led to a greater abundance of ALW mass. Heterogeneous or aqueous-phase reactions were favored with increasing ALW, promoting the formation of more oxidized SOA in liquid particles. At RH > 70%, the significant increase in $k_{org}$ (~14%) compensated for the negative effect of decreased $frac_{org}$ on the total hygroscopicity contributed by organics ($k_{org} \cdot frac_{org}$), leading to a stable $k_{org} \cdot frac_{org}$ with increasing RH (Figure 6c and Figure S15). This, in turn, coordinated with the increased $frac_{inorg}$, resulting in the second enhancement of

$k_{total}$. As a result, phase transition accompanied by increasing ALW mass triggered a
noticeable enhancement in $k_{total}$ with a mean value of 23% during haze episodes. The
enhanced water uptake ability of aerosols is expected to contribute more ALW under
elevated RH conditions, further facilitating the secondary aerosol formation and
deteriorating air quality. These results indicate that the establishment of positive
feedback loops between ALW and secondary aerosol formation was triggered by phase
transition during haze episodes.
**4 Conclusion and atmospheric implications**
Our findings revealed that particles predominantly exist as semi-solid or solid during
clean winter days with RH below 30%. However, non-liquid to liquid phase transition
occurred when the ALW mass fraction surpassed 15% (dry mass) at transition RH
thresholds ranging from 40% to 60%. Additionally, we observed a consistent pattern in
the non-liquid to liquid phase transition during haze formation, as manifested by both
particle-rebound fraction and bulk-phase viscosity measurements. Specifically, the
decrease in $f$ from >0.8 to 0 corresponded to a viscosity transition ranging from ~$10^8$
Pa s to ~$10^2$ Pa s. With the incorporation of $k_{total}$ into $T_g$ calculation for ambient OA, we
found that the characteristic of $T_g/T$ versus ALW/NR-PM$_1$ agrees well with our field
observations. This finding offers insights into the effectiveness of ALW/NR-PM$_1$ as an
indicator for quantifying the moisture-induced phase transition capacity of atmospheric
particles. Furthermore, incorporating overall particle hygroscopicity into the $T_g$
calculation may potentially enhance OA viscosity simulations, especially in regions
with a high proportion of inorganic salts under humid conditions. During haze episodes,
SOR and NOR rapidly increased through phase transition and increased ALW by 48%
and 11%, respectively, resulting in noticeable increases in SIA. The presence of
abundant ALW, favored by elevated RH and higher proportion of SIA, facilitates
heterogeneous and aqueous processes in liquid particles, leading to a substantial
increase in the formation of secondary organic aerosols and elevated aerosol oxidation.
As a result, the overall hygroscopicity parameters exhibit a substantial enhancement
with a mean value of 23%.
In our previous studies, we have revealed the positive feedback loops between ALW
and anthropogenic SIA at elevated RH levels during haze formation (Wu et al.,
2018;Wang et al., 2020b). The contribution of abundant ALW to SOA production has
also been reported in various regions with active anthropogenic emissions, such as the
Po Valley in Italy, southeastern U.S., and Beijing, China (Carlton and Turpin,
2013;Hodas et al., 2014;Xu et al., 2017). However, we observed that secondary
transformation of SIA and SOA was significantly enhanced after phase transition with
higher ALW mass during the observation. Our findings indicate that the secondary
aerosol formation could be impeded on non-liquid particles due to limited mass transfer
between gas and particle phase for relevant reaction components (Ravishankara,
1997;Shiraiwa et al., 2011;Abbatt et al., 2012;Ma et al., 2022), whereas it is facilitated
in liquid particles. It is therefore recommended that non-liquid to liquid phase transition
may be considered to be the kick-off for the positive feedback loops between ALW and
secondary aerosol formation during haze events. This can be further supported by the
case studies for varying polluted episodes, where episodes with phase transition
generally exhibit higher secondary transformation rate of secondary aerosols compared
to episodes without phase transition (Figure S16 and Text S7). This mechanism is
expected to gain significance in other regions with abundant anthropogenic emissions
and high background RH during haze formation.
**Author contributions**
X.X.Y.M. and Z.J.W. conceived the study. X.X.Y.M. conducted the experiments,
analyzed the experimental data, and wrote the manuscript with contributions from
Z.J.W., M.J.S., J.Y.L. and M.H. J.C.C. participated in the offline experiments and data
analysis. Y.T.Q. and T.M.Z. participated in the field experiments and conducted the
filter sampling.

## Funding

This work was supported by the National Natural Science Foundation of China (Grant No. 42375093) and the Fine Particle Research Initiative in East Asia Considering National Differences (FRIEND) Project through the National Research Foundation of Korea (NRF: 2020M3G1A1114537) funded by the Ministry of Science and ICT, Korea.

## Data availability

The data presented in this article can be accessed through the corresponding author Zhijun Wu (zhijunwu@pku.edu.cn).

## Acknowledgments

We sincerely thank our two referees for their valuable comments and constructive suggestions to improve the scientific and rigorous nature of our manuscript. We gratefully acknowledge the assistance of Wenfei Zhu for the technical support on Q-ACSM running and instrument calibration during the campaign.

## Competing interests

The authors declare that they have no conflict of interest.

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

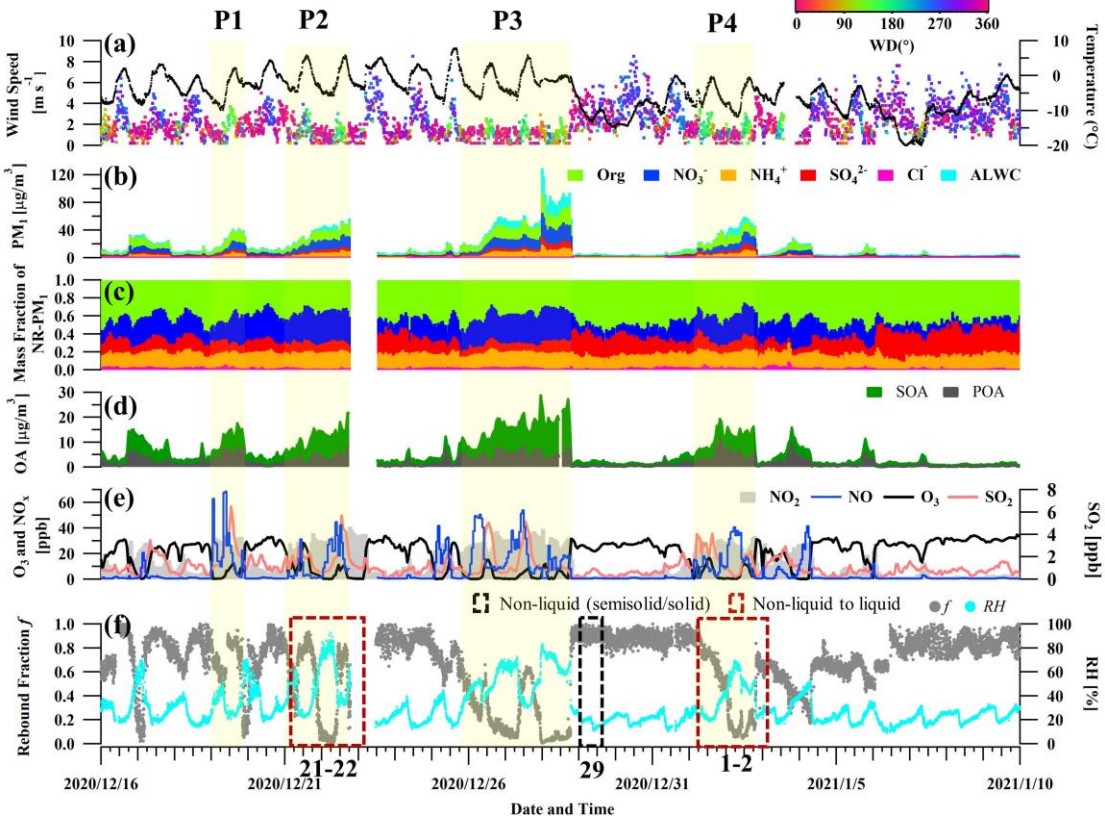


Figure 1. Time series of (a) wind speed (WS), wind direction (WD) and temperatures, (b) mass concentration of NR-PM$_1$ and ALW, (c) mass contribution of NR-PM$_1$, (d) mass concentrations of SOA and POA, (e) concentrations of gas pollutants (NO$_2$, NO, O$_3$, and SO$_2$), (f) rebound fraction and ambient RH during the field campaign. In panel (f), the black (red) frame with dashed line represents the non-liquid state (transition from non-liquid to liquid state) of bulk PM$_{2.5}$ droplets based on off-line viscosity measurement using poke-and-flow technique (Song et al., 2022).

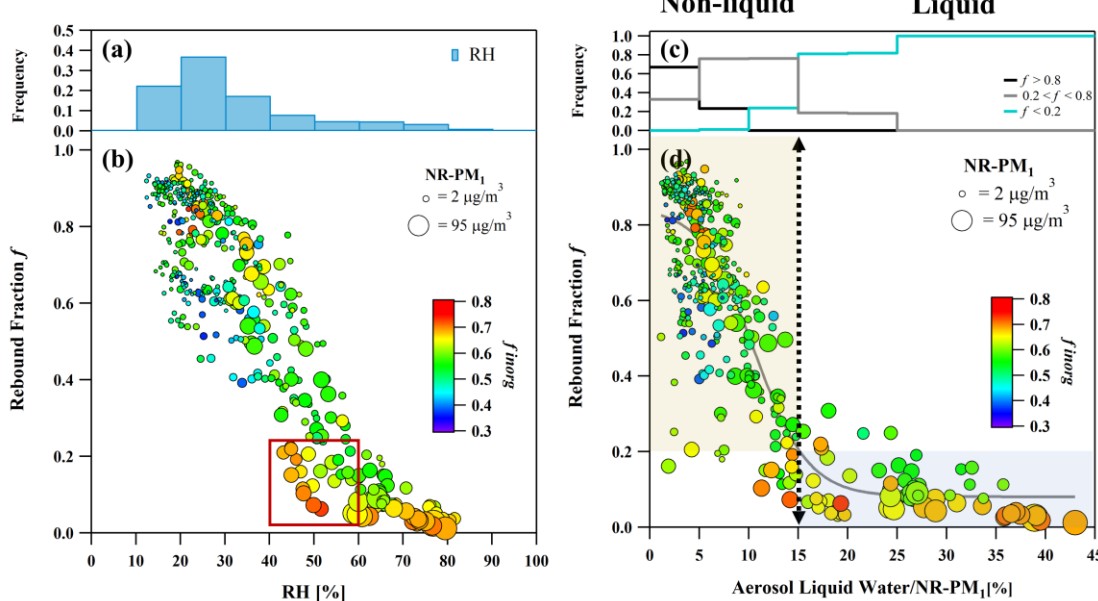


Figure 2. The frequency distribution of ambient RH in each RH bin (a) and the
frequency distribution of each $f$ interval in each ALW/NR-PM$_1$ bin (c). Rebound
fraction $f$ as a function of ambient RH (b) and ALW/NR-PM$_1$ (d) during the observation.
In panel (b) and (d), the scatter points are colored by $f_{inorg}$ in NR-PM$_1$ and the point size
is scaled by NR-PM$_1$ mass concentration. The yellow and blue shadow represent the
non-liquid and liquid phase, respectively.

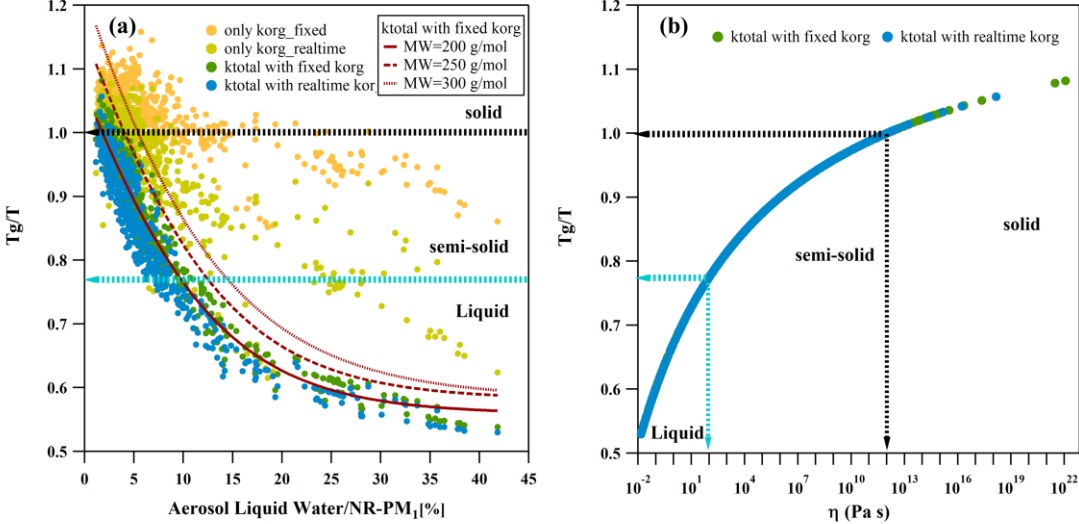

Figure 3. Characteristic relations between $T_g/T$ and ALW/NR-PM$_1$ (a) and $T_g/T$ as a function of predicted viscosity $\eta$ (b) of organic aerosols under ambient conditions. In panel (a), the red curves, which employ sigmoid fitting, represent variations in average molecular weights of OA used for $T_g$ calculation in consideration of the total hygroscopicity of the particles. The characteristics of the particle-phase state are delineated by arrows and dashed lines.

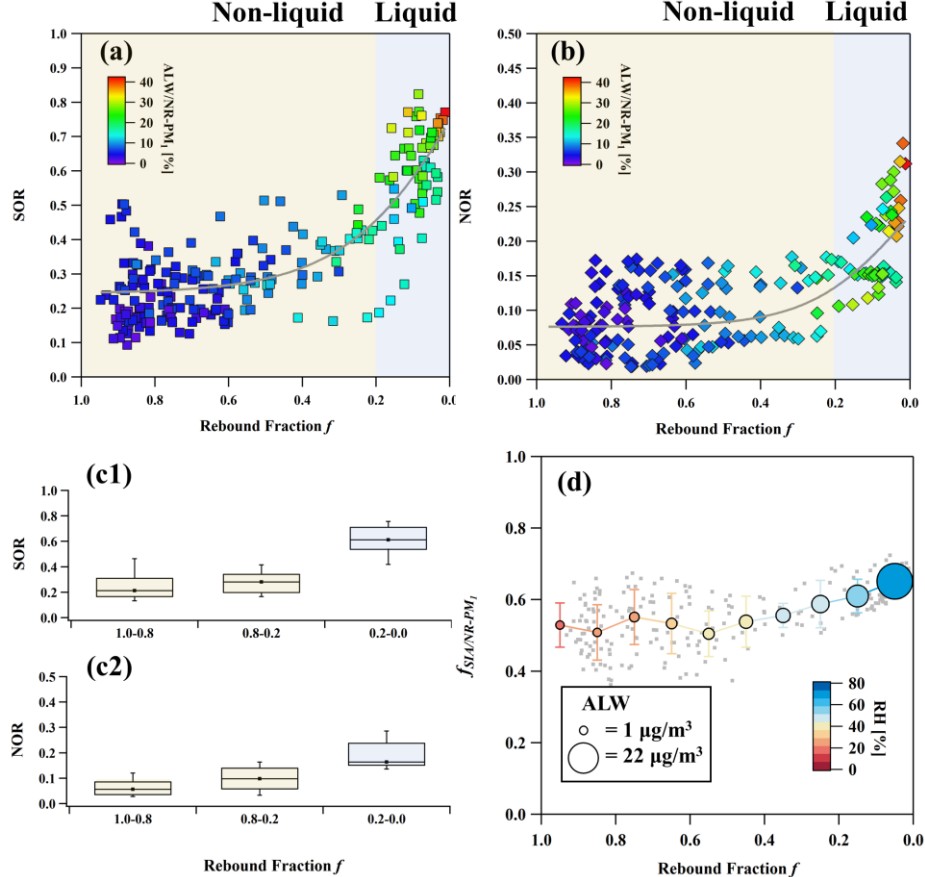

898

Figure 4. SOR and NOR as a function of $f$ (a, b), relationship between SOR or NOR and three phase transition level (c1, c2), and the mass fraction of SIA in NR-PM$_1$ as a function of $f$ during haze episodes (d). Non-liquid particles are marked by yellow shadows and liquid particles are marked by blue shadows. In panel (a) and (b), the scatter points are colored by ALW/NR-PM$_1$ and the trend lines are obtained by sigmoid fitting. In panel (c1) and (c2), the box plots show 10th, 25th, median,75th and 90th percentiles. In panel (d), RH is indicated by color, and ALW mass concentration is indicated by the size of the circle. The error bars show one standard deviation.


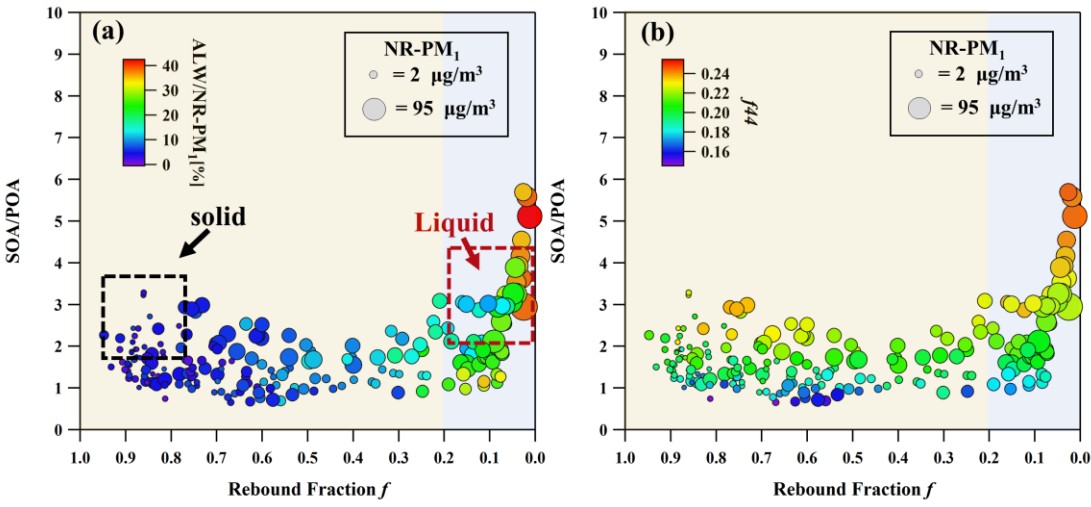

Figure 5. The relationship between SOA/POA and particle rebound fraction $f$ for phase
transition (a) and oxidation degree (b) during haze episodes. Non-liquid particles are
marked by yellow shadows and liquid particles are marked by blue shadows. The circles
are colored by ALW/NR-PM$_1$ and $f_{44}$ to represent water uptake capacity and particle
oxidation degree in panel (a) and panel (b), respectively. The sizes of the circles are
scaled to NR-PM$_1$ mass concentrations. The black (red) frame with dashed line
represent the off-line viscosity measurement results using poke-and-flow technique
corresponding to Figure 1.

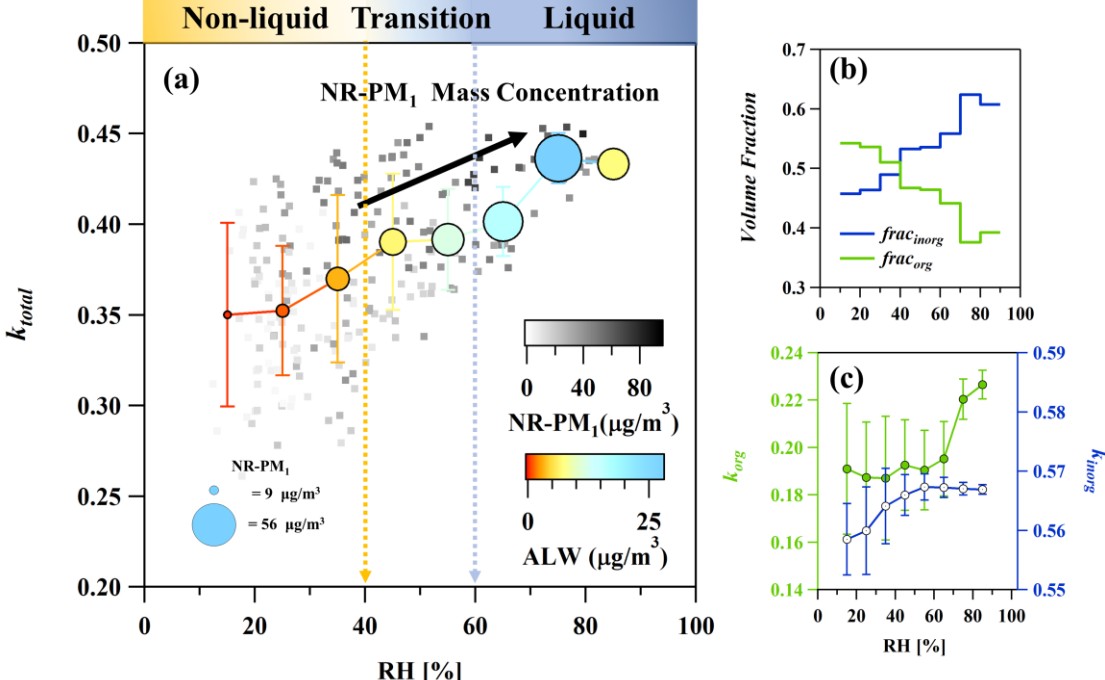


Figure 6. The overall hygroscopicity of particles (a), average volume fraction (b) and
hygroscopicity (c) of inorganics and organics as a function of RH during haze episodes.
$k_{total}$ was calculated using real-time $k_{org}$. Particles in different phase state condition,
including non-liquid, phase transition from non-liquid to liquid, and liquid, are visually
distinguished through a gradual color change from yellow to blue, which correlates with
RH. In panel (a), the scatter points are colored by NR-PM$_1$ mass concentrations and
averaged in each RH bin. Averaged ALW and NR-PM$_1$ mass concentrations are
indicated by color and the size of the circle, respectively. The error bars show one
standard deviation.