# Peer review of "Particle Phase State and Aerosol Liquid Water Greatly"

_EGUsphere, 2023_

## Author Comment (AC1)

*We are grateful to the editor and referees for their careful reading and constructive suggestions that substantially help to raise the quality of our manuscript. Below we address each of the comments listed in blue font. Our answer is listed in black font and revised text is listed in green font. The number of lines in our answers is based on the revised manuscript, and the amendments were marked with a highlight in the revised version.*

**Referee #1:**

This work describes measurements done in Beijing to understand phase transition during haze events using particle rebound and poke flow bulk viscosity measurements. The findings indicate that increased RH during haze events leads to particle phase transition from a semi-solid or solid to liquid phase state. This is shown to be due to increased inorganic fractions as well as uptake of more hygroscopic organics at elevated RH, both of which are promoted by uninhibited bulk diffusion in the liquid phase state. Overall, the level of detail and explanations of the results in this work are great and the limitations of interpretations of the results are clear. This work fits well within the scope of ACP and I would recommend it for publication once a few comments are addressed.

We appreciate the referee's review and very positive evaluation of our work. The comments were responded point-by-point in the following contents, and the manuscript was revised. We have made direct responses and substantial revisions which we believe properly address the referee's concerns.

**General comments:**
1. A limitation of using particle rebound fractions for ambient samples seems to be that a rebound fraction of 0.5 could indicate all your particles are semi-solid or 50% are solid and 50% are liquid and thus depends heavily on the mixing state of those ambient particles. Is there a way to validate that the particles sampled were internally mixed? Do the bulk viscosity measurements help to address this limitation?
We thank the referee for this comment.
We agree that using particle rebound measurement has such limitations on ambient samples since their mixing state may have impact on the number fractions of particle rebound. Therefore, to minimize this influence, we selected 300 nm mono-disperse particles by a Differential Mobility Analyzer for this measurement. As mentioned in Section 2.1 and 3.2, the measured *f* was representative of accumulation mode particles that dominated the mass concentration of submicron particles. In fact, validating the internally mixed state, even for mono-disperse particles, is challenging since the size-resolved chemical information is normally be inaccessible. However, the aerosols are supposed to be externally mixed during the clean period but turned to be internally

mixed during haze events as summarized by Peng et al (2021). Thus, the particle rebound measurement in this study fitted the aim furthest to show the phase state variation during the haze formation. The bulk viscosity measurement captured the phase information for water soluble components in $PM_{2.5}$ filter samples, so that internally mixed chemical composition was established. However, this approach cannot validate the mixing state of the measured mono-disperse particles. It should be noted that we achieved good agreement between the online measurement of particle rebound and offline viscosity results obtained by poke and flow technique, demonstrating the feasibility of this approach. We have now established a new measurement approach based on particle rebound for ambient particles and conducted a field campaign, incorporating synchronous Hygroscopic Tandem Differential Mobility Analyzer (HTDMA) measurements to obtain hygroscopic mode information of specific mono-disperse particles. We aim to elucidate these influences in our subsequent studies.

Jianfei Peng, Min Hu, Dongjie Shang, Zhijun Wu, Zhuofei Du, Tianyi Tan, Yanan Wang, Fang Zhang, and Renyi Zhang Environmental Science & Technology 2021 55 (4), 2189-2207 DOI: 10.1021/acs.est.0c07204

2. It is repeatedly suggested that multiphase chemistry is responsible for the increased oxidation of organics and the increased fraction of inorganics in particles at increased RH during haze events. However, could increased partitioning of these species due to higher aerosol-associated water not also explain these observations without any actual chemistry?

We thank the referee for this comment. Currently, delineating the contributions of different pathways to the secondary aerosol formation remains a significant challenge in high-resolution field measurements. Feng et al., (2023) utilized the isotopes measurement method to explore the formation pathways and contributions of nitrates. It was found that both gas-particle partitioning processes and heterogeneous reactions are enhanced during pollution episodes. In studies of secondary organic aerosols, the presence of abundant liquid water was suggested to facilitate the transfer of glyoxal from the gas phase to the particle phase, including both the partitioning process and the reactive uptake process, thereby increasing the mass concentration of organic aerosols or even the oxidation state (Volkamer et al., 2007;Hodas et al., 2014). In addition to liquid water, several other factors, such as temperature, oxidant levels, ionic strength, etc., also influence the formation pathways of these secondary products, adding considerable complexity to the research. Therefore, in our study, we have taken a cautious approach in interpreting the observed increase in ALW and its association with enhanced secondary aerosol formation rates via phase transition, and to focus on understanding the potential impact of phase state variations on secondary aerosol

formation. We agree that an increase in gas to particle partitioning of water-soluble compounds could also enhance the contribution of inorganics in particles or the oxidation state of organics. Therefore, we have discussed the potential influence on such pathway in the Section 3.3 and Section 3.4. We have noticed that this possibility was inadvertently omitted in the abstract, and it has now been rephrased in the revised manuscript:

Line 27-35:

"The presence of abundant ALW, favored by elevated RH and higher proportion of SIA, facilitates the partitioning of water-soluble compounds from gas to particle phase, as well as heterogeneous and aqueous processes in liquid particles. This leads to a substantial increase in the formation of secondary organic aerosols and elevated aerosol oxidation."

Feng, X., Chen, Y., Chen, S., Peng, Y., Liu, Z., Jiang, M., Feng, Y., Wang, L., Li, L., and Chen, J.: Dominant Contribution of NO3 Radical to NO3– Formation during Heavy Haze Episodes: Insights from High-Time Resolution of Dual Isotopes Δ17O and δ18O, Environmental Science & Technology, 57, 20726-20735, 10.1021/acs.est.3c07590, 2023.

Volkamer, R., F. San Martini, L. T. Molina, D. Salcedo, J. L. Jimenez, and M. J. Molina (2007), A missing sink for gas-phase glyoxal in Mexico City: Formation of secondary organic aerosol, Geophys. Res. Lett., 34, L19807, doi:10.1029/2007GL030752.

Angell, C. A.: Relaxation in Liquids, Polymers and Plastic Crystals - Strong Fragile Patterns and Problems, J Non-Cryst Solids, 131, 13-31, 1991.

Hodas, N., Sullivan, A. P., Skog, K., Keutsch, F. N., Collett, J. L., Decesari, S., Facchini, M. C., Carlton, A. G., Laaksonen, A., and Turpin, B. J.: Aerosol Liquid Water Driven by Anthropogenic Nitrate: Implications for Lifetimes of Water-Soluble Organic Gases and Potential for Secondary Organic Aerosol Formation, Environmental Science & Technology, 48, 11127-11136, 10.1021/es5025096, 2014.

3. This work shows more oxidized SOA in liquid particles with higher ALW. Can the authors comment on how this might affect the efficacy of commonly used parameterizations for OA viscosity, particularly those in Shiraiwa et al (2017) and DeRieux et al (2018) that would predict higher viscosities with higher levels of organic oxidation? Are these parameterizations still consistent with the results shown here if a composition dependent hygroscopicity parameter is used when calculating total aerosol viscosity?

We thank the insightful comments made by the referee.

Tg parameterizations are commonly used to predict the OA viscosity with considering the mixture of organics and organics-associated water. However, ambient aerosols

usually comprise high levels of inorganic and organic components. In our study, inorganics accounted for about 88% of the total ALW mass during the observation (with fixed $k_{org}$), implying that inorganics play a more significant role in driving the moisture-induced phase transition. It is a great point to consider the composition- dependent hygroscopicity parameter when calculating the viscosity of ambient organic aerosols through $T_g$ parameterizations. We followed your suggestion and employed the overall particle hygroscopicity, including both inorganics- and organics- associated water, rather than focusing solely on organics for calculating the $T_g$ of OA components under ambient conditions. The parameterization method proposed by Shiraiwa et al (2017) was employed. Unfortunately, we are unable to utilize the other two Tg parameterizations based on the volatility and elemental composition of organics, because of the lack of necessary input data. The common goal of viscosity studies is to accurately simulate and predict the evolution of viscosity in models. At the same time, concise and elegant parameterizations are also the pursuit of scientific researchers. We found that the characteristic of $T_g$/T versus ALW/NR-PM1 agrees well with our field observations with the incorporation of overall particle hygroscopicity into $T_g$ calculation for ambient OA. This finding suggests that a composition-dependent hygroscopicity parameter may be considered in regions characterized by higher mass concentrations of inorganics, particularly under humid conditions.

Combining the comments of the two referees, we chose to be cautious in comparing the calculation parts to our observation results, and to focus on the driving factors of the phase transition behaviors of these ambient particles observed in our study. This section was thoroughly rephrased in the revised manuscript and in the supplement:

[revised manuscript text omitted]

Added TexS6 in the supplement:
"The glass transition temperatures of organic aerosols under dry conditions ($T_{g,org}$) are calculated by the parametrization based on their molecular weight (M) and O:C ratio as below (Shiraiwa et al., 2017):

$$T_g = A + BM + CM^2 + D(O:C) + EM(O:C),  \qquad (S1)$$

Where A=-21.57 K, B=1.51 K mol g$^{-1}$, C=-1.7×10$^{-3}$ K mol$^2$ g$^{-2}$, D=131.4 K and E=-0.25 K mol g$^{-1}$, respectively. Here, we adopted an average molecular weight of 200 g mol$^{-1}$, as used in previous studies (Williams et al., 2010;Shen et al., 2018). O:C ratio was calculated by the parametrization of O:C=0.079+4.31×$f_{44}$ (Canagaratna et al., 2015).
The glass transition temperatures of organic-water mixtures (indicate the organic

aerosols under ambient conditions) can be simulated based on the Gordon-Taylor equation (Gordon and Taylor, 1952):

$$T_g(w_{org}) = \frac{(1-w_{org})T_{g,w} + \frac{1}{k_{GT}}w_{org}T_{g,org}}{(1-w_{org}) + \frac{1}{k_{GT}}w_{org}}, \qquad (S2)$$

where $w_{org}$ is the mass fraction of OA in the organic-water mixture, $T_{g,w}$ is the glass transition temperature of pure water (136 K), $T_{g,org}$ is the glass transition temperature of OA under dry conditions, and $k_{GT}$ is the Gordon-Taylor constant which is assumed to be 2.5 (Koop et al., 2011). The mass concentration of water in the organic-water mixture is commonly treated as organics-associated water, assuming an externally mixed phase of ambient particles (Shiraiwa et al., 2017;DeRieux et al., 2018;Li et al., 2020). Considering the significant contribution of inorganics to the total ALW mass of ambient aerosols, and the optically undetected liquid-liquid phase separation under staged dehydration of filter-based Beijing PM$_{2.5}$ droplets (Song et al., 2022), we assume in this study that OA particles are internally mixed with inorganic compounds such as sulfate and nitrate. Therefore, we adopted an overall particle hygroscopicity approach to calculate the total ALW mass in the organic-water mixture with the consideration of fixed $k_{org}$ and variable $k_{org}$. The mass concentration of water of ambient aerosols are calculated, as detailed in Section 2.2.

Then, the viscosity $\eta$ of ambient OA can be estimated by applying the Vogel–Tammann–Fulcher (VTF) equation (Angell, 1991):

$$\eta = \eta_\infty e^{\frac{T_0 D}{T-T_0}}, \qquad (S3)$$

where $\eta_\infty$ is the viscosity at infinite temperature (10$^{-5}$ Pa s; Angell, 1991), D is the fragility parameter, which is assumed to be 10 (DeRieux et al., 2018), and $T_0$ is the Vogel temperature calculated as $T_0 = \frac{39.17 T_g}{D+39.17}$."

**Specific comments:**

1. A fixed korg was used in this study and the supplement shows how a real-time korg greatly impacts the organic-associated water content. Were any sensitivity studies done on the fixed korg to see how the specific value of the fixed korg affects the results?

We thank the referee for this comment.

Calculations of ALW have predominantly considered the contribution of inorganic salts in numerous studies. However, the calculations demonstrate that the organics-associated ALW cannot be ignored regardless of whether $k_{org}$ remains fixed or varies with the degree of oxidation. In this study, we highlight the significant role of aerosol chemical composition and ambient RH in the evolution of particle-phase state during haze development. Acknowledging the plasticizing effect of water in particle-phase transition, we take the organics-associated ALW into account.

Our findings reveal that using only the content of ALW as a parameter does not adequately represent particle phase transitions. Instead, the proportion of liquid water in dry NR-PM$_1$ plays a more significant role. Therefore, although the contribution of organics to the overall ALW mass may be affected by variations in its hygroscopicity, its impact on ALW/NR-PM$_1$ is minor (see Figure S11).

Additionally, the relationship established in this study between ALW/NR-PM$_1$ and phase transitions from non-liquid to liquid has shown good performance. We also observe that the contribution of inorganic salts to the overall ALW is significantly higher than that of organic matter, indicating that inorganic salts remain the primary factor driving phase transition as RH increases during pollution events. Thus, we evaluate the frequency distribution of the three rebound fraction intervals in each ALW/NR-PM$_1$ bins for different ALW calculations by only inorganics, fixed $k_{org}$, and real-time $k_{org}$. As expected, the frequency distribution showed no obvious change among these three situations. The choice of a fixed $k_{org}$ in this study does not affect our conclusions, and a lower value of 0.06 is acceptable.

We strongly concur with the opinions of the two reviewers and have incorporated a sensitivity analysis in the corresponding sections of the revised manuscript as well as in the supplement:

Manuscript (L169-172):

"In recognition of ALW's plasticizing effect on particle-phase state, the potential impacts of whether the hygroscopicity value of organics is fixed or varies in real-time on phase transition has been discussed in Section 3.2."

Manuscript (L269-285):

"It should be noted that calculations of ALW in this study have considered inorganic salts and organics. Acknowledging that the hygroscopicity of organics, characterized by either a fixed $k_{org}$ or varying in real-time, affects the calculation of ALW mass, a sensitivity analysis examining its impact on the phase transition threshold (ALW/NR-PM$_1$) is presented in Figure S11. There is no denying that the contribution of inorganic salts to ALW remains predominant, with their contribution to ALW being ~88% (a fixed $k_{org}$ of 0.06) and ~73.5% (real-time $k_{org}$) on average during the observation (Figure S4), indicating that the impact of different ALW calculations on ALW/NR-PM$_1$ values was not significant. As expected, the frequency distribution of these three $f$ intervals showed no obvious change for ALW calculations by inorganics, fixed $k_{org}$, and real-time $k_{org}$ at the whole ALW/NR-PM$_1$ range. Although the frequency of $f < 0.2$ changed from ~0.8 to 0.5 at ALW/NR-PM$_1$ = 15-20% when shifting to real-time $k_{org}$, the frequency remained higher than 0.5 and approached 1.0 with larger ALW/NR-PM$_1$

values. This indicates that while the varying $k_{org}$ impacts the number of data points within ALW/NR-PM$_1$ bins of 15-20%, it does not affect the overall transition trend. As a result, the impacts of different ALW calculation on ALW/NR-PM$_1$ were not significant, and the phase transition threshold of 15% remains valid."

Supplement (Figure S11):

[Figure]

Added Figure S11. The frequency distribution of three *f* intervals ( *f* > 0.8 and *0.2 <f < 0.8* and *f* < 0.2) in each ALW/NR-PM$_1$ bins (a and b) and *f* as a function of ALW/NR-PM$_1$ using three different methods of ALW calculation (c).

2. Line 364: At this point it's been awhile since ktotal was introduced and it may be helpful to remind readers what it is here.

We followed the comment and modified the sentence:

"In Figure 5a, the relationship between the overall particle hygroscopicity (k$_{total}$) and RH is displayed."

3.   Fig 5c: Why does kinorg level off at high RH, while korg continues to increase?

We thank the referee for this comment.

Considering the variability in the composition of inorganics and organics, we used either real-time korg characterized by the parametrization of $k_{org} = 1.04 \times f_{44} -$

0.02 and varied kinorg characterized by weighted volume fractions in inorganics as detailed in Section 2.2. The increasing trend of korg was attributed to the elevated oxidation degree of organics at higher RH conditions during the haze events. This was attributed to a clear increasing trend in $f_{44}$ observed after phase transition and at higher ALW mass conditions. kinorg was determined by changing volume fraction of inorganic species ($k_{NH_4NO_3}=0.58$ and $k_{(NH_4)_2SO_4}=0.48$). With increasing RH, kinorg showed an upward trend due to increased nitrate contribution in total inorganics during the haze periods. However, nitrate contribution remained stable at RH>60% as well as higher particulate mass concentrations (see Figure. S5). This explains the leveling off of kinorg at higher RH values.

4. Fig S4 does not have a legend

We followed the comment and modified the figure as shown below:

[Figure]

Figure S4. Modified Figure. S5

---

## Author Comment (AC2)

*We are grateful to the editor and referees for their careful reading and constructive suggestions that substantially help to raise the quality of our manuscript. Below we address each of the comments listed in blue font. Our answer is listed in black font and revised text is listed in green font. The number of lines in our answers is based on the revised manuscript, and the amendments were marked with a highlight in the revised version.*

**Referee #2:**

Meng et al. conducted particle rebound measurements and inferred the phase state of fine particles. They also analyzed the mass concentrations of chemical compositions in particles measured by ACSM and calculated the aerosol liquid water (ALW) content. They showed that the particle phase transition is a key factor initiating the positive feedback loops between ALW and secondary aerosol formation during haze episodes over the North China Plain. The manuscript is well written and the observation data is carefully analyzed and clearly presented. As the particle phase state measurements and analysis are still limited in East Asia, this study has significance understanding the role of particle phase state in aerosol multiphase chemistry and secondary aerosol formation in hazy days in megacities. I recommend the publication of this study after the following comments could be addressed.

We appreciate the referee's affirmation and comments on our work. The comments were responded point-by-point in the following contents, and the manuscript was completely revised. We believe the referee's concerns have been addressed.

**General comments:**
1. Particle phase state is related with particle chemical composition and RH, which was detailed analyzed in this study. However, besides chemical composition and RH, ambient temperature also affects the particle phase state (Koop et al., 2011). From Fig. S12 I found the temperatures between clear days and polluted episodes can be over 10 ℃ different. I suggest the authors add analysis on the relationship between temperature and particle phase state and discuss the potential effects of temperature on multiphase chemistry and gas-particle partitioning.

We thank the insightful observation made by the referee.

Our findings reveal that the impact of temperature on particle rebound fraction was not as significant as that of RH in our study (See Figure S13). Specifically, a decrease in temperature leads to an increase in viscosity, while an increase in RH reduces viscosity, attributed to the plasticizing effect of water (Koop et al., 2011). Although average temperatures changed from -10℃ to ~0℃ between clean and polluted episodes, environmental RH exhibited a wider variation, ranging from approximately 20% to

80%. Consequently, RH is presumed to have a more prominent influence on phase transition than temperature in the near-surface atmosphere. Nonetheless, we acknowledge that temperature could potentially affect multiphase chemistry and gas-particle partitioning, as higher temperatures generally correlate with increased rates of gas or particle phase diffusion coefficients (Tang et al., 2014; Li and Shiraiwa, 2019). We have incorporated this additional discussion in the revised manuscript and the supplement:

Line 308-315 in the manuscript:
"In addition to RH and aerosol compositions, environmental temperature also plays a significant role in determining the phase state (Koop et al., 2011;Shiraiwa et al., 2017;Petters et al., 2019). A reduction in temperature results in higher viscosity, whereas a rise in RH leads to a decrease in viscosity, attributed to the plasticizing effect of water (Koop et al., 2011). Although the relationship between $f$ and temperature is not strongly evident as that of RH in this study (Figure S13), it's observed that a greater number of data points exhibited near 0.9 under low RH conditions (<30%), suggesting higher viscosity at colder temperatures (< -10 ℃) than warmer scenarios."

Added Figure S13:

[Figure]

Added Figure S13. Particle rebound fraction dependency of environmental temperature. The point size is scaled by NR-PM$_1$ mass concentration. The points are colored by environmental RH.

Line 398-404 in the manuscript:
"One should note that, the average environmental temperature during pollution

episodes increased to approximately 0°C, in contrast to the -10°C recorded during clean periods. The rise in ambient temperature typically enhances the diffusivity of atmospheric reactive molecules in both the gas and particle phases (Tang et al., 2014;Shiraiwa et al., 2011;Li and Shiraiwa, 2019). This, in turn, may potentially influence the heterogeneous or liquid-phase reactions, and even the gas-particle partitioning of semi-volatile compounds."

Koop, T., Bookhold, J., Shiraiwa, M., and Poschl, U.: Glass transition and phase state of organic compounds: dependency on molecular properties and implications for secondary organic aerosols in the atmosphere, Phys. Chem. Chem. Phys., 13, 19238-19255, 10.1039/C1CP22617G, 2011.

Tang, M. J., Cox, R. A., and Kalberer, M.: Compilation and evaluation of gas phase diffusion coefficients of reactive trace gases in the atmosphere: volume 1. Inorganic compounds, Atmos. Chem. Phys., 14, 9233-9247, 10.5194/acp-14-9233-2014, 2014.

Li, Y., and Shiraiwa, M.: Timescales of secondary organic aerosols to reach equilibrium at various temperatures and relative humidities, Atmos. Chem. Phys., 19, 5959-5971, 10.5194/acp-19-5959-2019, 2019.

**Specific comments:**

1.  Line 19: This study focused on effects of phase transition and particulate water on secondary aerosol formation, and the particle growth was not particularly investigated. I suggest change "winter particulate growth".

We thank the referee for this suggestion. The sentences were rephrased:

"This study provides valuable insights into the significance of particle-phase transition and aerosol liquid water (ALW) in particle mass growth during winter."

2.  Line 115-120: I am not an expert in experiments, but I am curious how long it takes for the impactor RH to be equal to the ambient RH? Did the rebounded particles reach equilibrium with the impactor RH during the measurement? This would be helpful to convince the readers that the measured phase state indeed is the phase state at the ambient RH.

We thank the referee for this comment.

We agree that the time for the impactor RH to be equal to the ambient RH is important for our measurement and should be clarified here. We added more detailed description in the revised version of our manuscript and the supplement:

Line 120-129 in the manuscript:

"The measured impactor RH rapidly reached the ambient RH within 1 second, exhibiting a mean absolute error of 0.03. This swift regulation time is attributed to the real-time feedback in the RH control system, coupled with the typically modest

fluctuations in ambient RH. Particles with a diameter of 300 nm, as selected for our study, rapidly achieved equilibrium in the humidification process, since the timescale for water diffusion into these particles is approximately 1 second, shorter than their residence time of about 3 seconds within the system. This is detailed in Text S2 and illustrated in Figure S1. Such conditions ensure that the measured particle rebound are representative and accurate at the ambient RH."

Added Text S2 in the supplement:

"In this study, the characteristic half-time for water diffusion ($\tau_{1/2}$) is employed to give basic information about the timescales for water molecules diffusion into particles in the RH adjustment system. $\tau_{1/2}$ is given by Seinfeld and Pandis (2006) as $\tau_{1/2} = \frac{r^2}{\pi^2 D \ln(2)}$, where r is the radius of the particles and D is the water diffusion coefficient. It should be noted that the diffusion timescale within particles is calculated at a constant water activity, meaning that D remains constant as well. An RH of 80% represents the nearly highest RH conditions during our observation, potentially indicating an upper level of the humidification process. For particles with a diameter of 300 nm, $\tau_{1/2}$ was approximately 1.03 seconds. This was calculated using a constant water activity (80% RH) at room temperatures, with the water diffusion coefficient being $10^{-10}$ m$^2$ s$^{-1}$ (Koop et al., 2011). Particles with diameter smaller than 300 nm have even shorter water diffusion timescales (e.g., 0.46 seconds for 200 nm particles; 0.11 seconds for 100 nm particles). This is consistent with the laboratory study by Price et al. (2014), which quantified water diffusion in high-viscosity aerosols and determined that water diffusion timescales are less than 1 second for particles with a radius smaller than 250 nm at room temperature. The measured particles were initially dried to ~30% RH before entering the sampling line. However, these particles rapidly reached ambient RH conditions and passed through the RH adjustment system, which had a tested residence time of about 3.3 seconds using a highly humid flow pulse as shown in Figure S1. Given that the timescales for water diffusion into particles were much shorter than their residence time, it is presumed that particles rapidly reached equilibrium with the impactor RH during the measurement."

Added Figure S1 in the supplement:

[Figure]

Modified Figure S1. Time after pulse for RH adjustment system by using humid flow go through the flow path.

3.  Line 128: Change "organic" to "organics" or "organic aerosol".
Modified (Line 136).

4.  Line 150: Delete "be" in "it should be note that" and check this all through the manuscript, e.g. Line 213, 274 and 289.
We thank the referee for this comment. We changed it to "it should be noted that" in the revised manuscript (Line 159, Line 225, Line 302 and Line 367).

5.  Line 156-157: The calculated fixed korg of 0.06 seems at the lowest end of the reported range in winter Beijing and lower than the predicted real-time korg. As korg affects the aerosol water which affects the phase state and further other results of this study, I agree with the first reviewer that sensitivity calculations should be done to evaluate the impacts of korg on the results of this study.
We thank the referee for this comment.
Calculations of ALW have predominantly considered the contribution of inorganic salts in numerous studies. However, the calculations demonstrate that the organics-associated ALW cannot be ignored regardless of whether korg remains fixed or varies with the degree of oxidation. In this study, we highlight the significant role of aerosol chemical composition and ambient RH in the evolution of particle-phase state during haze development. Acknowledging the plasticizing effect of water in particle-phase transition, we take the organics-associated ALW into account.

Our findings reveal that using only the content of ALW as a parameter does not adequately represent particle phase transitions. Instead, the proportion of liquid water in dry NR-PM$_1$ plays a more significant role. Therefore, although the contribution of organics to the overall ALW mass may be affected by variations in its hygroscopicity, its impact on ALW/NR-PM$_1$ is minor (see Figure S11).

Additionally, the relationship established in this study between $ALW/NR\text{-}PM_1$ and phase transitions from non-liquid to liquid has shown good performance. We also observe that the contribution of inorganic salts to the overall ALW is significantly higher than that of organic matter, indicating that inorganic salts remain the primary factor driving phase transition as RH increases during pollution events. Thus, we evaluate the frequency distribution of the three rebound fraction intervals in each $ALW/NR\text{-}PM_1$ bins for different ALW calculations by only inorganics, fixed $k_{org}$, and real-time $k_{org}$. As expected, the frequency distribution showed no obvious change among these three situations. The choice of a fixed $k_{org}$ in this study does not affect our conclusions, and a lower value of 0.06 is acceptable.

We strongly concur with the opinions of the two reviewers and have added a sensitivity analysis in the corresponding sections of the revised manuscript and the supplement:

Manuscript (L169-172):
"In recognition of ALW's plasticizing effect on particle-phase state, the potential impacts of whether the hygroscopicity value of organics is fixed or varies in real-time on phase transition has been discussed in Section 3.2."

Manuscript (L269-285):
"It should be noted that calculations of ALW in this study have considered inorganic salts and organics. Acknowledging that the hygroscopicity of organics, characterized by either a fixed $k_{org}$ or varying in real-time, affects the calculation of ALW mass, a sensitivity analysis examining its impact on the phase transition threshold ($ALW/NR\text{-}PM_1$) is presented in Figure S11. There is no denying that the contribution of inorganic salts to ALW remains predominant, with their contribution to ALW being ~88% (a fixed $k_{org}$ of 0.06) and ~73.5% (real-time $k_{org}$) on average during the observation (Figure S4), indicating that the impact of different ALW calculations on $ALW/NR\text{-}PM_1$ values was not significant. As expected, the frequency distribution of these three $f$ intervals showed no obvious change for ALW calculations by inorganics, fixed $k_{org}$, and real-time $k_{org}$ at the whole $ALW/NR\text{-}PM_1$ range. Although the frequency of $f < 0.2$ changed from ~0.8 to 0.5 at $ALW/NR\text{-}PM_1$ = 15-20% when shifting to real-time $k_{org}$, the frequency remained higher than 0.5 and approached 1.0 with larger $ALW/NR\text{-}PM_1$ values. This indicates that while the varying $k_{org}$ impacts the number of data points within $ALW/NR\text{-}PM_1$ bins of 15-20%, it does not affect the overall transition trend. As a result, the impacts of different ALW calculation on $ALW/NR\text{-}PM_1$ were not significant, and the phase transition threshold of 15% remains valid."

Supplement (Figure S11):

[Figure]

Added Figure S11. The frequency distribution of three $f$ intervals ( $f > 0.8$ and $0.2 < f < 0.8$ and $f < 0.2$) in each ALW/NR-PM$_1$ bins (a and b) and $f$ as a function of ALW/NR-PM$_1$ using three different methods of ALW calculation (c).

6.  I agree with the General comment 3 of the first reviewer that the dependence of viscosity on oxidation state should be discussed. Dette et al. (2014), Koop et al. (2011), Li et al. (2020) and Saukko et al. (2012) are helpful for this discussion.

Thanks for your comments.

In this study, we did not explore the connection between oxidation state and viscosity, as there was no available online measurement for quantitative viscosity values to discuss such an influence, particularly in terms of ambient aerosols comprising high levels of inorganic and organic components. We could discuss such connection using a simple parameterization of the glass transition temperature ($T_g$) of OA components based on their molar mass and O:C ratio. However, this approach would only be applicable to OA, not the whole particle. This is due to the fact that $T_g$ calculations for OA commonly consider the mixture of organics and organics-associated water. In our study, inorganics accounted for about 88% of the total ALW mass during the observation, implying that inorganics play a more significant role in driving the moisture-induced phase transition. Therefore, our approach involved considering the overall particle hygroscopicity, including both inorganics- and organics-associated

water, rather than focusing solely on organics for calculating the $T_g$ of OA under ambient conditions. Although this approach was simplified by setting the average molecular weight of SOA at a fixed value of 200 g/mol, we considered various influencing factors of viscosity, such as oxidation state, chemical composition, RH and temperature in our revised manuscript.

Combining the comments of the two referees, we chose to be cautious in comparing the calculation parts to our observation results, and to focus on the driving factors of the phase transition behaviors of these ambient particles observed in our study. This section was thoroughly rephrased in the revised manuscript and in the supplement:

[revised manuscript text omitted]

Added TexS6 in the supplement:
"The glass transition temperatures of organic aerosols under dry conditions ($T_{g,org}$) are calculated by the parametrization based on their molecular weight (M) and O:C ratio as below (Shiraiwa et al., 2017):

$$T_g = A + BM + CM^2 + D(O:C) + EM(O:C), \qquad (S1)$$

Where A=-21.57 K, B=1.51 K mol g$^{-1}$, C=-1.7×10$^{-3}$ K mol$^2$ g$^{-2}$, D=131.4 K and E=-0.25 K mol g$^{-1}$, respectively. Here, we adopted an average molecular weight of 200 g mol$^{-1}$, as used in previous studies (Williams et al., 2010;Shen et al., 2018). O:C ratio was calculated by the parametrization of O:C=0.079+4.31×$f_{44}$ (Canagaratna et al., 2015).

The glass transition temperatures of organic-water mixtures (indicate the organic aerosols under ambient conditions) can be simulated based on the Gordon-Taylor equation (Gordon and Taylor, 1952):

$$T_g(w_{org}) = \frac{(1-w_{org})T_{g,w} + \frac{1}{k_{GT}}w_{org}T_{g,org}}{(1-w_{org}) + \frac{1}{k_{GT}}w_{org}}, \qquad (S2)$$

where $w_{org}$ is the mass fraction of OA in the organic-water mixture, $T_{g,w}$ is the glass transition temperature of pure water (136 K), $T_{g,org}$ is the glass transition temperature of OA under dry conditions, and $k_{GT}$ is the Gordon-Taylor constant which is assumed to be 2.5 (Koop et al., 2011). The mass concentration of water in the organic-water mixture is commonly treated as organics-associated water, assuming an externally mixed phase of ambient particles (Shiraiwa et al., 2017;DeRieux et al., 2018;Li et al., 2020). Considering the significant contribution of inorganics to the total ALW mass of ambient aerosols, and the optically undetected liquid-liquid phase separation under staged dehydration of filter-based Beijing PM$_{2.5}$ droplets (Song et al., 2022), we assume

in this study that OA particles are internally mixed with inorganic compounds such as sulfate and nitrate. Therefore, we adopted an overall particle hygroscopicity approach to calculate the total ALW mass in the organic-water mixture with the consideration of fixed $k_{org}$ and variable $k_{org}$. The mass concentration of water of ambient aerosols are calculated, as detailed in Section 2.2.

Then, the viscosity η of ambient OA can be estimated by applying the Vogel–Tammann–Fulcher (VTF) equation (Angell, 1991):

$$\eta = \eta_\infty e^{\frac{T_0 D}{T-T_0}}, \qquad (S3)$$

where $\eta_\infty$ is the viscosity at infinite temperature ($10^{-5}$ Pa s; Angell, 1991), D is the fragility parameter, which is assumed to be 10 (DeRieux et al., 2018), and $T_0$ is the Vogel temperature calculated as $T_0 = \frac{39.17 T_g}{D+39.17}$."

7.    Line 261: The authors found several points with ALW/NR-PM1 < 5% and NR-PM1 > 30 µg/m3 exhibited lower rebound fraction (f < 0.4) in Figure 2d and Figure S9, and they gave two possible reasons based on analyzing the ratio of ALW/NR-PM1. Why you chose ALW/NR-PM1 instead of ALW to interpret the results? If ALW is used for the interpretation, would the explanation be different?

We thank the referee for this comment.

In this study, we find that using ALW mass as a parameter is not entirely adequate for representing the moisture induced phase transition. For example, we found that particle rebound fraction decreased with increasing ALW mass, reaching $f < 0.2$ when ALW exceeded 10 µg/m$^3$ (see Figure S10). However, we observed that some particles with higher $f_{inorg}$ transitioned to a liquid state at 5 µg/m$^3$ < ALW < 10 µg/m$^3$. Additionally, there was a considerable variation in the rebound fraction among particles with the same ALW mass. Instead, ALW/NR-PM$_1$ show good correlation with rebound fraction. This indicates that it is the mass fraction of ALW, rather than the absolute ALW mass, that significantly influences particle-phase transition. This is the reason why we chose ALW/NR-PM$_1$ instead of ALW to interpret the Figure 2d. The two possible reasons discussed here are not affected by different forms of the figure, as more detailed information is available in Figure S12.

We modified the sentence to improve the clarity of the statement (Line 263-264) and added one Figure in the Supplement (Figure S10):

Line 263-264 in the manuscript:

"… indicates that particles mostly convert to liquid when the mass fraction of ALW surpasses a certain threshold during haze formation, rather than the absolute ALW mass (Figure S10)."

[Figure]

Added Figure S10. Particle rebound fraction as a function of aerosol liquid water content during the whole observation in Beijing. The scatter points are colored by the mass fraction of inorganic matter ($f_{inorg}$) in NR-PM$_1$.

8.  Line 368: I think 56 μg/m3 is for NR-PM1 instead of ALW.

We thank the referee for this comment and modified the sentence (Line 452):

"… and further rose to 0.43 with an average maximum NR-PM$_1$ value of 56 μg/m$^3$ when RH reached 70-80%."

9.  Line 373: Why do the mass concentrations of NR-PM1 and ALW decrease in the highest RH bin in Figure 5a?

We thank the referee for this comment.

The observed lower mass concentrations of NR-PM$_1$ and ALW data points in the highest RH bin (>80%) are related to the P2 haze episode, one haze episode characterized by the highest RH value during the whole campaign. However, the averaged mass concentration of NR-PM$_1$ for P2 (33.7 μg/m$^3$) was lower than that of P3 (43.7 μg/m$^3$). Additionally, ALW mass was determined by aerosol chemical composition, mass loading, and environmental conditions (RH and Temperature). As shown in Figure S16, P2 and P3 exhibit similar mass fractions of NR-PM$_1$ composition. And the mean temperature of P2 and P3 did not show much differences. Although P2 has several higher RH values than P3, resulting in these data points being categorized in the highest RH bin, the mean RH of P3 was actually higher than that of P2. Thus, higher ALW mass was favored in P3. This led to a decrease in ALW in highest RH bin in Figure 6a. Even though, we noticed that these observed data points stay in a higher $k_{total}$ in Figure 6a.